# Gene Expression Profile of Uterine Leiomyoma from Women Exposed to Different Air Pollution Levels in Metropolitan Cities of Sao Paulo, Brazil

**DOI:** 10.3390/ijms24032431

**Published:** 2023-01-26

**Authors:** Laura Gonzalez dos Anjos, Bruna Cristine de Almeida, Edmund Chada Baracat, Ayman Al-Hendy, Qiwei Yang, Katia Candido Carvalho

**Affiliations:** 1Laboratório de Ginecologia Estrutural e Molecular (LIM 58), Disciplina de Ginecologia, Departamento de Obstetricia e Ginecologia, Hospital das Clinicas da Faculdade de Medicina da Universidade de Sao Paulo (HCFMUSP), São Paulo 05403-010, Brazil; 2Department of Obstetrics and Gynecology, University of Chicago, Chicago, IL 60637, USA

**Keywords:** uterine leiomyoma, air pollution, gene expression, real-time PCR, tumorigenesis, fertility

## Abstract

Leiomyomas (LMs) are the most frequent uterine benign tumors, representing the leading cause of hysterectomy indications worldwide. They are highly associated with women’s reproductive complications, and endocrine disruptors may influence their etiology. In this sense, air pollution represents a relevant hormonal disruptor that acts on key signaling pathways, resulting in tumor development and infertility. Our goal was to evaluate submucosal LM samples from patients living in the metropolitan and Sao Paulo city regions, focusing on genes involved in tumor development and infertility features. Twenty-four patients were selected based on their region of residence and clinical information availability. Several genes were differentially expressed between women living in metropolitan areas and Sao Paulo city. Significant associations were observed between *BCL-2*, *DVL1*, *FGFR3*, and *WNT5b* downregulation and contraceptive use in the samples from women living in Sao Paulo city. *ESR1* and *HHAT* downregulation was associated with ethnicity. *WNT5b* and *GREM* were associated with LM treatment and related pathologies, respectively. In the samples from women living in other cities of the metropolitan region, abortion occurrence was associated with *BMP4* upregulation. Although further studies may be necessary, our results showed that air pollution exposure influences the expression of genes related to LM development and female reproductive features.

## 1. Introduction

Uterine leiomyomas (LMs), also known as uterine fibroids, are the most common benign tumors affecting the gynecological tracts of reproductive-age women [1,2,3]. Their clinical symptoms may include abnormal uterine bleeding, dysmenorrhea, and pelvic pain. They have a relevant impact on natural or in vitro fertility [4,5,6]. Recurrent abortion and infertility have also been associated with this condition [1,2,7]. Sunkara et al. [8] demonstrated that even intramural LMs, which do not induce distortions in the uterine cavity, are associated with increased pregnancy loss rates and reduced fecundity. Lately, an association between submucosal LMs and subfertility was observed in a meta-analysis and systematic review [6]. The authors described several reproductive effects of these tumors, such as the disruption of physiological myometrial motility, the destabilization of the pelvic anatomy, and a deleterious endometrial inflammatory milieu [6].

Molecular studies have shown a variety of genes that might be involved in LM development, such as *MED12* [9] and *HMGA2* [10]. These genes are frequently found mutated in uterine LM. Dysregulation in the *WNT*, *SHH*, and *TGF-β* signaling pathways was also identified [11,12]. Additionally, LM growth may occur through estrogen and progesterone activities that activate the pathways involved in cell proliferation and differentiation [13,14] or through the *WNT/β-catenin* pathway, which is frequently associated with endometrial receptivity [15]. This pathway, beyond targeting *MED12*, is associated with increased levels of *TGF-β3*, acting as a mediator of *BMP2* production and *HOXA10* expression [15]. *HOXA10* belongs to the homeobox (Hox) family of genes, which are critical to functional endometrial development during the menstrual cycle and endometrial receptivity [16]. *HOXA10* and *HOXA11* induce downstream factors, influencing endometrial receptivity by activating or repressing their target genes [17].

An increasing amount of evidence supports the hypothesis that LM arises from abnormal stem cells in the myometrial smooth muscle compartment of the uterus [18,19]. Such cells seem to originate from a point mutation in key genes and may present complex histologic features that interfere directly with the diagnosis [20]. Endocrine disruptors and racial or ethnic factors may contribute to genetic alterations in myometrial stem cells [18,19,20,21]. Additionally, these cells are more frequent in Afro-descendant women with LMs than in Caucasians without them [21].

Over the last decade, chronic exposure to ambient air pollution (AP) has become increasingly recognized as an important risk factor for LM development [22]. The serious implications and health impairments arising from these adverse conditions may be perpetuated for subsequent generations [22,23,24,25]. Adverse environmental exposure, both in utero or in early life, can lead to metabolic and epigenetic changes that culminate in LM development and reproductive complications in adulthood and subsequent generations [24,25].

AP involves different pollutants present in the atmosphere. Some studies have identified AP as an endocrine disruptor, acting in several reproductive organs [21]. Additionally, particulate matter (PM) is a class of pollutants comprising a combination of small-sized particles and gaseous components (organic chemicals, smoke, soot, sulfates, nitrates, acidic components, dust particles, soil, and others) [20,21,22]. A recent study demonstrated that chronic exposure to PM_2.5_ was associated with the incidence of clinically symptomatic LMs [23]. In addition, chronic exposure to PM_2.5_ led to endocrine disturbance, menstrual irregularity, and infertility, as described in several studies [20,21,22,23,24].

Although some recent epidemiological and experimental discoveries have advanced our current understanding of LM risk and etiology, little is known about the molecules directly associated with its origin and its effects on women’s fertility. Concerning AP’s effects on LM development and its association with women’s infertility, there are few studies in the literature describing their potential molecular correlation. Here, we performed a gene expression profile study, focusing on the signaling pathways involved in tumor development and their association with fertility and endometrial receptivity. LM samples were selected from patients who lived in Carapicuiba, Cotia, Diadema, Franco da Rocha, Itapevi, Osasco, Riberão Pires, and Vargem Grande Paulista, areas within the metropolitan region of Sao Paulo (MRSP) and the Sao Paulo (SP) capital city.

## 2. Results

### 2.1. Air Quality and Patients’ Clinicopathological Features

AP records were analyzed to compare the air quality between Sao Paulo city and MRSP. Data analysis showed no significant alteration in the levels of pollutants during the period included in the present study (2002–2020). According to the established pattern of air quality, the highest number of days exceeding the standard levels of air quality (52 days) was observed in 2020. In the last 10 years (2011–2020), the floating average of unfavorable days for particle dispersion by year was between 23 and 56. During the last 18 years, no parameter exceeded the alert-limit established values, according to the average values based on Environmental Company of Sao Paulo State—CETESB reports (Table 1). Noteworthy is the fact that several MRSP cities presented higher levels of AR than Sao Paulo. In 2020, the PM_2.5_ level was 29 in Campinas city and 113 in Ribeirao Preto, while in SP, in the University City and Marginal Tiete regions, the levels were 66 and 64, respectively. Sao Caetano city belongs to the MRSP and has higher levels of pollution (68) compared to Sao Paulo. The main difference between the metropolitan region and Sao Paulo city was not in the levels of pollutants, but in the number of days that exceeded the standard levels. The evolution curves for PM_10_, PM_2.5_, smoke, CO, and O_3_ all presented a decreasing profile for the last 18 years. 

From the available clinical data pertaining to G1 (MRSP) and G2 (Sao Paulo city) women evaluated in the present study, we observed an average age of 41.9 ± 6.71 years (ranging from 27 to 54 years old). The average menarche age was 13.1 ± 1.9 (ranging from 11 to 20 years old), and the average body mass index (BMI) was 27.8 ± 5.8 (ranging from 18.4 to 40.0). Patients presented uterine volumes ranging from 126 to 2.632 cm^3^ (646.8 ± 557.3). All these clinical features were homogeneous according to the comparison between G1 and G2 (Table 2).

The clinical feature comparison between G1 and G2 showed that 60% (9) of the oldest patients were in G2. Only 25% (6) of the patients were Afro-descendant women, and 83% (5) of these were in G2. Patients in G2 showed higher rates of nulliparity, smoking, and abortion (67%), and 56% had undergone a hysterectomy. Less than 50% (10) of the patients had received pharmacological treatment against LM; of these, 80% were in G2. In addition, 50% of the patients in both groups presented associated pathologies, mainly diabetes and hypertension (Table 3).

Although there were no significant differences in clinical features between the groups, we performed a comparison of clinical data and the expression profiles of genes related to key pathways involved in tumor development and genes associated with fertility and endometrial receptivity in these samples.

### 2.2. Gene Expression and Signaling Pathway Analyses

Using the open-array system (Taqman^®^ detection method), we assessed the expression profile of 112 genes (including six endogenous controls) involved in several signaling pathways (*WNT*, *SHH*, *MAPK*, and others) and biological processes (cell apoptosis, proliferation, and others). Thirty-two genes were not detected in the RG, resulting in 75 genes observed using a non-hierarchical method (Figure 1). Two samples were excluded from the analysis due to their housekeeping-genes amplification profile (Ct ≥ 30). Figure 1 shows several genes with differences in expression between the LM samples (G1 and G2) and the RG (MM samples). The heatmap shows the gene expression profile for each LM sample compared to the average RG expression. This analysis provided non-normalized and non-hierarchized results that allowed us to observe the general expression profile of the genes individually for each sample.

After data normalization, the genes with differential expression among the groups were selected for the next analyses. Considering the relative quantification values (RQs), we found 41 genes in G1 and 50 in G2 with differential expression (Figure 2a). To enhance the biological significance and simplify the validation of the expression results, the cutoff values were increased to <−2 for downregulation and >2 for upregulation.

Figure 2b shows six differentially expressed genes (DEGs) in G1 (*BMP7*, *DISP2*, *FZD5*, *PRL*, and *PTCH1* (upregulated) and *JUN* (downregulated)) and seven (*BMP7*, *CCND1*, *FZD5*, *GREM1*, *PRL*, *SFRP4*, and *SMO*) in G2, compared to RG. As our main interest was genes with a differential profile of expression in women with a longer exposure to high levels of pollutants, the ratio of expression in G2 compared with G1 was also calculated (Figure 2c). This analysis was performed using <−2 and >2 as the cutoff values, and we found seven upregulated genes (*BMP4*, *CTNNBP1*, *DISP1*, *DVL1*, *ERBB4*, *PRLR*, and *WNT5b*) and three downregulated genes (*LEF1*, *PGR*, and *PTEN*) in G2 (Figure 2c).

The correlation analyses of the gene expression values showed significant correlation between several genes (Figure 3). Four relevant genes, i.e., *WNT2*, *GSK3B*, *MYC*, and *ESR1*, presented the highest correlations with other genes. *WNT2* showed a significant correlation with *SFRP1*, *PGR*, *PTEN*, *FOXO3a*, and *DISP2*. *GSK3B* was correlated with *TP53*, *TLE1*, *MYC*, *FZD2*, *FOXO3a*, *DISP2,* and *DVL1*. *MYC* showed a correlation with *LATS2*, *LEF1*, *SFRP1,* and *DISP1*. In addition, *ESR1* showed a significant correlation with *WNT2*, *TLE1*, *SFRP1,* and *PTEN*. Figure 3 shows the Spearman correlation coefficients for the genes that presented statistical significance (*p*-values between 0.001 and 0.05).

The association between the gene expression profile and the available clinical data from the patients was assessed. Significant associations were observed between a lack of contraceptive use by patients from G2 and the downregulation of *BCL-2* (*p* = 0.0151), *DVL1* (*p* = 0.0208), *FGFR3* (*p* = 0.0194), and *HHAT* (*p* = 0.0194), and several genes (*GSK3B*, *DISP1*, *FZD2*, *PTEN*, *TP53*, *MAPK1*, *LATS2*, *SFRP1*, and *SLC2A3*) presented a trend toward statistical significance for this variable. Afro-descendant women from G2 also presented the downregulation of *ESR1* (*p* = 0.0424) and *HHAT* (*p* = 0.0160). Moreover, the downregulation of *GREM1* showed an association with the occurrence of associated pathologies (diabetes and hypertension).

In G1, the uterine volume was associated with *ERS1* upregulation (*p* = 0.0205), and abortion occurrence was related to *BMP4* upregulation (*p* = 0.0342). Appendix A show the performed analyses that presented at least one significant association.

### 2.3. Fertility- and Receptivity-Related Gene Analyses

Another analysis was performed using an array composed of 86 sequences of genes related to fertility and endometrial receptivity (Appendix A). Figure 4 shows the expression profile of 89 genes, including 86 target and 3 housekeeping genes. Seventeen samples were evaluated, four samples from G1, eight from G2, and five from RG. This analysis was performed for a lower number of samples because of the amount of RNA available and the software quality-control process, which found fault with the amplification or cDNA synthesis profiles of some samples. This clustergram performed the non-supervised hierarchical clustering of the entire dataset to display a heatmap with dendrograms indicating coregulated genes across groups or individual samples.

After gene normalization, we identified several DEGs between the RG and LM samples (G1 and G2). Fifteen genes were upregulated in G1, and 38 were downregulated in comparison with the RG (Figure 5a and Appendix A). Comparing G2 with RG, we observed that 17 genes were upregulated and 40 were downregulated (Figure 5b and Appendix A). When the comparison was performed between G2 and G1, the number of DEGs was lower than that previously found. Eight genes were upregulated (*SFRP4*, *KLRC1*, *ESR2*, *PRLR*, *TSHR*, *SLC16A6*, *C4BPA*, and *HLA-DOB*), and ten were downregulated (*OLFM4*, *DUOX1*, *CTNNA2*, *HABP2*, *CXCL14*, *MTNR1A*, *KISS1R*, *PENK*, *CALB2*, and *LIF*) in G2. In the scatter plots shown in Figure 5a–c, the gene expression profiles of G1 and G2 are plotted against each other to clearly visualize large differences. Figure 5d shows the data in a volcano plot, combing *p*-values with fold regulation changes, enabling the identification of those with both large and small expression changes.

The amplification profile of the 86 target genes can be seen in Figure 6. As a reference, the RG is always presented as the same value (=1). To assess the association between gene expression and patients´ clinical features, we included the 19 DEGs between G1 and G2 (*SFRP4*, *KLRC1*, *ESR2*, *PRLR*, *TSHR*, *SLC16A6*, *C4BPA*, and *HLA-DOB* (upregulated in G2) and *OLFM4*, *DUOX1*, *CTNNA2*, *HABP2*, *CXCL14*, *MTNR1A*, *KISS1R*, *PENK*, *CALB2*, and *LIF* (downregulated in G2)). Additionally, we included *HOXA10* and 11, *LITAF*, Mucins, and BMPs, due to their relationship with fertility and endometrial receptivity. However, their expression was lower than that of the other genes (>2 and −2). Among them, a significant association between gene expression and clinical data was observed only for *LITAF* and BMI (*p* = 0.040). A trend toward statistical significance was observed for *DUOX1*, *PCNA*, *HOXA 10*, and *CTNNA2*. Several assessed genes presented a correlation tendency with contraceptive use, BMI, abortion history, and ethnicity.

## 3. Discussion

The gene expression evaluation of submucosal LM samples from patients living in metropolitan areas and Sao Paulo city provided us with a molecular link between AP-exposure-related parameters and women’s health. The region of Sao Paulo city known as the expanded center has a concentration of urban infrastructure and services, in addition to a high road density and traffic volume [26]. Recently, in this region, there has been a decrease in industrial pollution, due to technological advances and migration. However, the cities located in the metropolitan region of Sao Paulo contain 47,000 industries and around 100,000 commercial establishments, which have contributed to a significant increase in AP and, consequently, public health issues [27].

It has already been well-described in the literature that AP can affect gene expression through DNA damage [28]. This phenomenon may affect reparative activity, making the uterine environment prone to chronic inflammation, which may result in LM development [29]. Although the etiology of LM remains unknown, the available data suggest that genetic alterations contribute to the benign transformation of origin cells. The human myometrial cells have similar characteristics to stem cells, which, due to the action of PM_2.5_ or O_3_, may become tumor-initiating cells (TICs) that cause specific genetic and epigenetic alterations [30].

Here, we showed that groups of women residing in the metropolitan region and Sao Paulo city presented different gene expression profiles in their LM samples. We evaluated the expression of genes involved in various signaling pathways (*WNT*, *SHH*, *MAPK,* and others) and biological processes (cellular apoptosis, proliferation, and others). The LM samples of women residing in Sao Paulo city presented the upregulation of *BMP4*, *CTNNB1*, *DISP1*, *DVL1*, *ERBB4*, *PRL,* and *WNT5b* and the downregulation of *LEF1*, *PGR,* and *PTEN* compared to the LM samples of women from other metropolitan regions.

Although women residing in metropolitan regions are exposed to high levels of AP for longer durations, different compounds may be found according to the polluting source. In megacities in developing nations, such as Sao Paulo, around 80% of the AP is attributed to vehicular emissions caused by older vehicles, poor vehicle maintenance, and low fuel quality [31]. Generally, vehicles are responsible for all most all CO, HC (hydrocarbon), and NOx emissions; half of all PM; and 42% of SOx, according to CETESB [32]. In cities with growing industrialization, there are already huge amounts of organic compounds in the air, as well as CO, HCs, and chemicals. These molecules can affect gene expression and change cell behaviors in different ways [33].

Garcia et al. showed that *BMP4* expression increased gradually depending on the histological type of uterine smooth muscle tissue, being lower in MM and higher in leiomyosarcomas (LMSs), a rare form of uterine cancer [12]. According to Zaitseva et al., CTNNB1 also has aberrant expression in LMs compared to matching MM. This gene is related to tumor progression in uterine diseases through epithelial–mesenchymal transition (EMT) mechanisms, which are mainly associated with endometrial disorders, including cancer [34]. Importantly, DISP1 plays an essential role in sonic hedgehog (SHH) patterning activities [35], and, just like BMP4, the protein expression of the SHH signaling components SMO, SUFU, GLI1, and GLI3 increases in LM and LMS compared to MM [12]. *HER4* (ErbB4) is part of the human epidermal growth factor receptor (EGFR) family. Its function and prognostic capacity are still unclear; however, its upregulation and the presence of mutations in solid tumors have already been demonstrated [36,37]. Yu et al. observed that *EGFR* expression is increased in LM compared to MM and may be related to cell transformation [38].

Another gene that appeared overexpressed in LM samples from residents of Sao Paulo city was *PRL*. Carbajo-García et al. performed an interaction analysis between DNA methylation and gene expression in LM compared to adjacent MM. The authors found hypomethylation and the upregulation of *PRL* in LM. Knowing that AP alters epigenetic markers, particularly DNA methylation (mDNA), the association of these factors may help to elucidate the pathogenesis of LM [39,40]. The Wnt pathway is altered in a large proportion of LM patients, and specific members, such as *DVL1* and *Wnt5b*, are aberrant in these tumors. This abnormal expression contributes to the progression of a high percentage of human cancers [11].

The loss of the expression of tumor suppressor *PTEN* can lead to a wide spectrum of human diseases. Its partial loss is enough to promote tumor development as well as contribute to cancer progression [41]. Depending on the context, *LEF1* loss can also be harmful, since in some cases, such as in T-cell acute lymphoblastic leukemia, this gene acts by repressing *MYC* transcription [42]. LMs are hormone-dependent tumors. Studies suggest that increased mitotic activity in the secretory phase of the cycle is associated with progesterone levels [43,44]. The physiological effects of progesterone are mediated by intracellular proteins called progesterone receptors (PRs) [45]. *PGR* is the gene responsible for encoding these receptors. In the pre-transcriptional stage, *PGR* loss is a consequence of promoter methylation, copy number loss, or even mutations. In addition, *PGR* mRNA is a direct target of several miRNAs, which may indirectly reduce *PGR* regulation [46].

For a better understanding of the harmful effects that pollution can have, specifically on fertility and endometrial receptivity, we focused on evaluating genes related to these processes and found differences between G1 and G2. Since AP has endocrine-disrupting properties, even at low concentrations, the alteration of *ESR2*, *PRLR*, *TSHR*, *DUOX1,* and *CALB2* may be related to regulatory mechanisms enacted by toxic particles. Urban AP, found in higher levels in Sao Paulo city, is also associated with inflammation [47]. There is evidence that the chronically inflammatory systemic immune profile is associated with endometrial receptivity alterations [48]. *KLRC1*, *C4BPB*, *HLA-DOB*, *OLFM4*, *CXCL14,* and *LEF* are genes involved in immune response regulation, such as complement activation, leukocyte migration, cytokine activity, and MHC complex activation [49]. The mechanisms of action of pollutants are multifactorial and depend on the tissue and exposure conditions [50]. In this context, gene expression evaluation may be useful to detect immune perturbations in uterine tissue.

In addition to the described alterations, it is important to highlight the *CTNNA2* downregulation in G1 and G2. This tumor suppressor gene is related to the regulation of muscle cell differentiation [49]. Although several factors are associated with tumor malignancy, this loss may contribute to the appearance of abnormal cells that look and behave differently to normal cells [51].

The clinical features of the women included in our study were also related to their gene expression profiles. Significant associations were observed between *BCL-2*, *DVL1*, *FGFR3*, and *WNT5b* downregulation and contraceptive use in the samples from women living in Sao Paulo (G2). It has already been suggested that LM growth and survival depend on differences in local factors and receptor patterns. Each tumor may respond differently to hormonal stimulation [52]. Qin et al. suggested that oral contraceptive use did not increase the probability of developing LMs, finding that the risk was reduced by 17% in those who had used oral contraceptives for more than 5 years [53]. Determining how the use of oral contraceptives, together with AP exposure, can affect women’s health may be challenging, as it involves epigenetic remodeling, susceptibility to polluting particle effects, and particularities regarding the general health status of each individual.

*ESR1* and *HHAT* downregulation was associated with ethnicity, while *WNT5b* and *GREM* were associated with LM treatment and pathologies, respectively. In the samples from women living in other cities of the metropolitan region, abortion occurrence was associated with *BMP4* upregulation. A wide range of studies have demonstrated that LM is more common in black women than in white women [21]. Many possible causes can be attributed to this phenomenon, such as higher levels of steroid hormones, ERα polymorphism, and even vitamin D deficiency. However, it is still not fully understood how ethnicity can interfere with LM etiology [54,55]. The aberrant expression of *BMP4* was associated with miscarriages in G1. Zhang et al. demonstrated that long-term maternal exposure to AP increases the likelihood of miscarriage, stillbirths, and birth defects [56]. Furthermore, this molecule might be related to the endometriosis etiology [57], a condition that can affect fertility and increases the risk of miscarriage by almost 80% [58].

In summary, LM is a highly heterogeneous tumor in both its genetics and histological characteristics as well as its clinical behavior. Although our focus was the gene expression screening method, we also tried to look for subgroups of relevant features for this disease’s initiation and progression, but the statistical significance was lost due to the limited number of patients in each group. Even though our study had limitations regarding the number of samples and regions from which women were selected, it was the first in Brazil to narrow down the relationship between AP, gene expression, and LM development. Building on this study, we will seek to select women from other regions of Sao Paulo state, such as the countryside and coastal regions, highlighting cities where the pollution index is considered lower. In this way, it will be possible to expand our data and fully ascertain the potential effects of atmospheric pollution on an increased risk of LM development.

Understanding the mechanisms enacted by polluting particles is highly necessary, as they have been proven to increase the risk of LM development. However, only a few studies have shown this association, such as that of Mahalingaiah et al., who conducted a prospective cohort study with women aged 25 and 42 years. This study revealed that mean cumulative PM_2.5_ concentrations of 15.2 µg/m^3^ were associated with an increased LM risk [22]. Lin et al. also showed this association in a 10-year cohort-based case–control study in Taiwanese women. This study was carried out in areas with high PM_2.5_ concentrations (33.3 µg/m^3^) and evaluated prolonged exposure effects [30]. In another recent study, Wesselink et al. conducted a prospective cohort study with 21,998 premenopausal black women. The authors demonstrated that ambient concentrations, particularly O_3_, were also associated with LM. These findings support the existence of unequal exposure to AP and higher rates of LM among black women [23]. As AP is an increasingly impactful factor in our society, future studies may contribute to the development of new targeted therapies and consequently improve women’s quality of life.

## 4. Materials and Methods

### 4.1. Case Selection

The present study was approved by the Institutional Research Ethics Committee of the Faculdade de Medicina da Universidade de Sao Paulo (numbers 0845/11 and 1815/13) and was conducted in accordance with the Declaration of Helsinki. Patient samples were obtained from the Department of Gynecology of Faculdade de Medicina da Universidade de Sao Paulo, Sao Paulo, Brazil. All patients were diagnosed with uterine LM and submitted to hysterectomy or myomectomy between 2012 and 2018, and all provided their informed consent for inclusion before they participated in the study. The clinical follow-up of patients continued for more than sixty months, and clinicopathological data were extracted from patient charts considering the information relevant to this study. Patients with any kind of infectious disease or cancer and those who provided no information about their area of residence were excluded from the study.

Additionally, we exclusively selected samples from patients who declared their precise region of residence, and a total of 24 submucosal LM samples were included in the molecular analyses. Samples were divided into two groups according to the patient’s city of residence: 10 samples from patients who lived in the MRSP (G1) and 14 samples from patients who lived in Sao Paulo city (G2) (Appendix A). Five samples of myometrium (MM) from patients in G1 were used as the reference group (RG) to identify DEGs between LM and MM.

### 4.2. Extraction of Total RNA, cDNA Synthesis, and qRT-PCR Assays

Total RNA from frozen samples was extracted using Trizol^®^ Reagent (Thermo Fisher Scientific, Waltham, MA, USA). Complementary DNA (cDNA) syntheses were performed using a High-Capacity kit (Thermo Fisher Scientific, Waltham, MA, USA) according to the manufacturer’s instructions with 2 µg of total RNA. Efficiency was evaluated by conventional PCR, using specific primers for the *ACTB* (actin beta) and *GAPDH* (glyceraldehyde-3-phosphate dehydrogenase) genes. We evaluated 106 genes related to several signaling pathways described as being involved in tumor development and 6 housekeeping genes (Appendix A). qRT-PCR assays were performed for 29 samples (10 from G1, 14 from G2, and 5 from RG) that presented a sufficient concentration and quality level for the use of this platform protocol. All reactions were performed in duplicate, using 1.2 µL of cDNA and 3.8 µL of TaqMan Gene Expression Master Mix (Thermo Fisher Scientific, Waltham, MA, USA). Reactions were carried out on the QuantStudio 12K Flex Open-Array Real-Time PCR System (Thermo Fisher Scientific, Waltham, MA, USA).

Open-array chips allow high throughput tracking experiments involving a small number of samples and reagents. Each chip contains 48 sub-arrays, and each sub-matrix has 64 true holes, totaling 3072 holes on the chip. The cycling conditions were applied, as recommended by the manufacturer. *ACTB*; *B2M* (beta-2-microglobulin); *GAPDH*; *GUSB* (glucuronidase beta); *HPRT1* (hypoxanthine phosphoribosyltransferase 1); and *RPLP0* (ribosomal protein, large, P0) were selected as housekeeping genes. Data were analyzed using Expression Suite software v1.0.3, applying the comparative Cτ (ΔΔCτ) method (Thermo Fisher Scientific, Waltham, MA, USA).

The second method used to analyze the expression profile of genes associated with fertility and endometrial receptivity was the Qiagen array plate, which comprised 89 sequences (Appendix A). The analyses were performed using 18 LM samples, including 5 from G1, 8 from G2, and 5 from RG. The total RNA of all samples was quantified by spectrophotometric NanoDrop (ND100, Thermo Fisher Scientific, Waltham, MA, USA), and the integrity profile was assessed by electrophoresis on 1% agarose gel. In the reverse-transcription reaction, a First Strand kit RT2 (QIAGEN—SAbiosciences Corporation, Hilden, Germany) was used to transcribe 1 µg of total RNA purified from each sample into cDNA according to the manufacturer’s instructions. The cDNA was subjected to qRT-PCR using a customized PCR array plate, including the sequences of selected genes in the RT2-profile PCR arrays (CLAH-27251A, QIAGEN—SAbiosciences Corporation, Hilden, Germany). Reactions were performed using a 7500 Real-time PCR System (Applied Biosystems, Foster City, CA, USA).

### 4.3. Air Quality Data Collection

The last 18 years of the Sao Paulo environmental data reports were consulted. Reports are available on the CETESB website (https://cetesb.sp.gov.br, accessed on 26 November 2020). Data such as total inhalable particles, PM_10_, PM_2.5_, ozone, smoke, and carbon monoxide were assessed. The values were presented according to the data available in the reports, as 8 or 24 h values and the annual arithmetic average (AAA). Other data were included as floating averages, because the specific daily or annual values were not available. During the evaluated time course (2002 to 2020), two different legislations were used to establish the standard parameters of air quality. The first was published in 1990 by the CONAMA institute (the Environment National Council), and the second by State law in 2013 (59.113/2013). Few alterations were included, but the total particulate material was separated into PM_10_ and PM_2.5_, and the limit levels of O_3_ and some pollutants were reduced.

### 4.4. Data Assessment and Statistical Analysis

To compare the differences in the expression of the selected genes among the patient groups (G1, G2, and the reference group RG), an analysis of the runs and amplification curves was performed for each of the samples for all evaluated genes, with the aid of Thermo Fisher Cloud software (https://www.thermofisher.com/account-center/cloud-signin-identifier.html, accessed on 15 June 2021). Based on these curves, samples or genes that did not present an adequate amplification profile were excluded according to the algorithm present in the software itself, as indicated by the manufacturer. In this step, 29 samples were evaluated (10 from G1, 14 from G2, and 5 from RG) in terms of their gene expression profile. The relative quantification values were calculated by the ΔΔCτ method using Thermo Fisher software (Expression Suite v1.0.3).

Gene expression results were provided as relative quantification (RQ) and fold-change (FC) values, and the cutoff signals were defined as FC ≤−2.0 and ≥2, compared to the selected control. The Qiagen array-based analysis of gene expression was performed using the SAbio-science online program, available at https://dataanalysis2.qiagen.com/pcr/analysis/52829, accessed on 18 October 2021. Data analysis was performed using the average, minimum, and maximum values; standard deviation; and absolute and relative frequencies. The average and standard deviation (average ± SD) were used to assess the patients’ age, and the chi-square test or Fisher’s exact test were used to evaluate associations between gene expression and clinicopathological features. All genes that did not present amplification in 100% of the samples were excluded from the analysis. Statistical analyses were performed using GraphPad Prism 5.0 statistical software (San Diego, CA, USA) and SPSS for Windows (SPSS Inc. Chicago, IL, USA). Statistical significance was accepted for *p*-values ≤ 0.05.

## 5. Conclusions

We found several genes dysregulated in LM samples from patients living in Sao Paulo state. All 10 DEGs found in the patients who lived in Sao Paulo city compared to the MRSP were associated with cell proliferation and tumor development (*BMP4, CTNNBP1, DISP1, DVL1, ERBB4, PRLR, WNT5b, LEF1, PGR*, and *PTEN*). Additionally, 18 genes related to fertility and receptivity were dysregulated in these women (*SFRP4*, *KLRC1*, *ESR2*, *PRLR*, *TSHR*, *SLC16A6*, *C4BPA, HLA-DOB, OLFM4, DUOX1*, *CTNNA2*, *HABP2*, *CXCL14*, *MTNR1A*, *KISS1R*, *PENK*, *CALB2,* and *LIF*). Several LM patients’ features and gene expressions were associated with AP exposure, but in women living in the MRSP, abortion occurrence was strongly associated with *BMP4* upregulation. Although further studies are needed to establish the fundamental role of AP exposure and the molecular mechanism regulation of the gene expression profile in LM, herein we described for the first time the relevant gene expression alterations associated with both AP exposure and LM risk in women living in Sao Paulo state, Brazil.

## Figures and Tables

**Figure 1 ijms-24-02431-f001:**
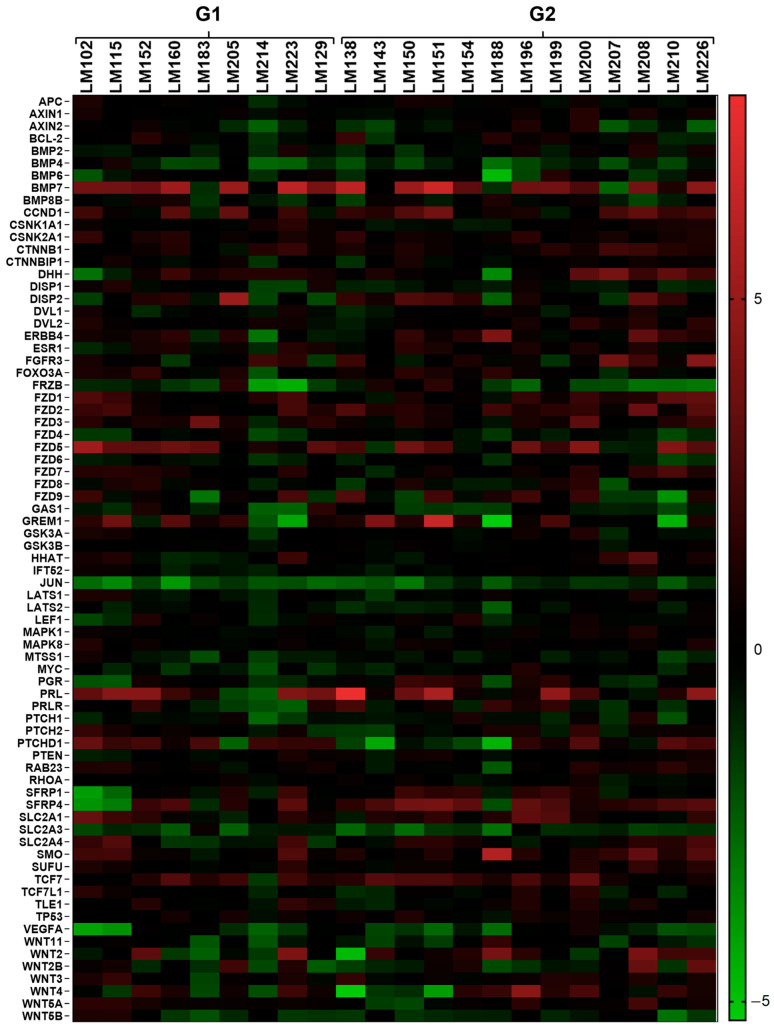
Heatmap showing the global gene expression profiles of the 75 sequences (out of 112 included in the chip) evaluated by qRT-PCR, using the open-array (TaqMan^®^ probes and primers) method. G1 = MRSP, G2 = SP, RG = myometrium from G1. Red indicates upregulation, green indicates downregulation, and black indicates no differences between target samples and RG. The color scale bar indicates the grade of significance according to the gene expression value (from −5 to 5) compared to the RG (MM).

**Figure 2 ijms-24-02431-f002:**
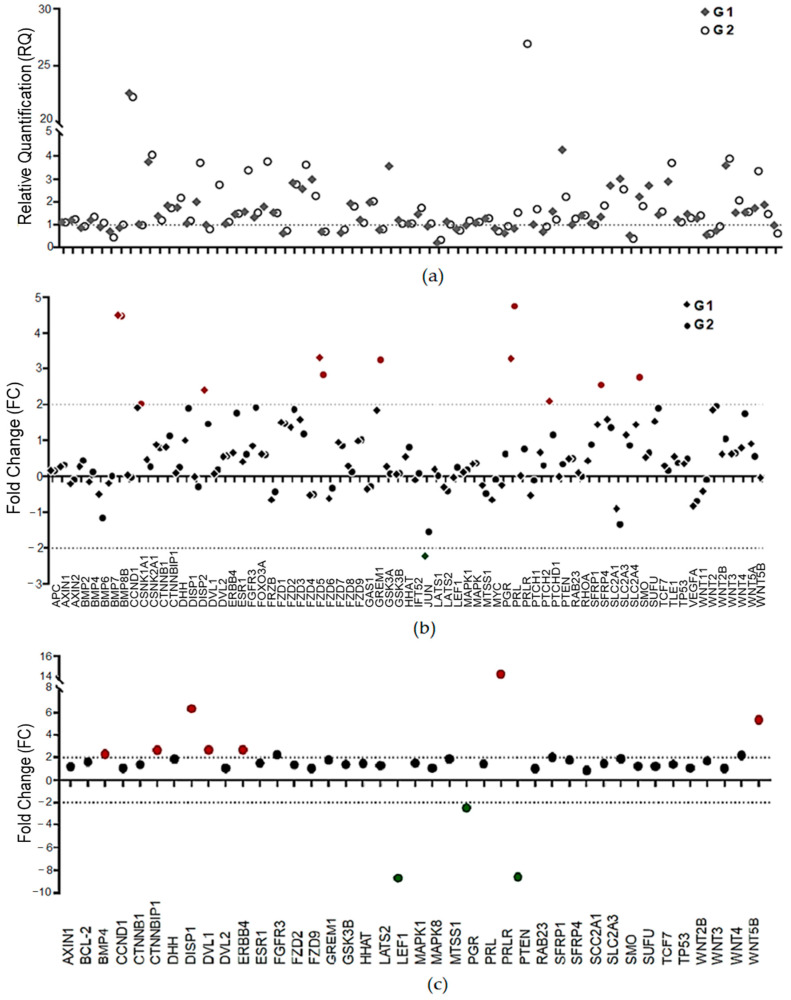
Gene expression profile determined by open-array assays (TaqMan^®^ method of detection). (**a**) Profile of relative quantification (RQ) of the 75 genes in LM samples (G1 and G2) compared to RG (reference samples = 1); (**b**) fold-change values (FC) of the genes in the G1 and G2 samples (cutoff >2 and <−2); (**c**) the ratio of expression values between 2 and 1 after normalization and the selection of 41 genes that presented high amplification efficiency (FC > 1) in all the LM samples. Red indicates upregulated genes, while green indicates downregulated genes.

**Figure 3 ijms-24-02431-f003:**
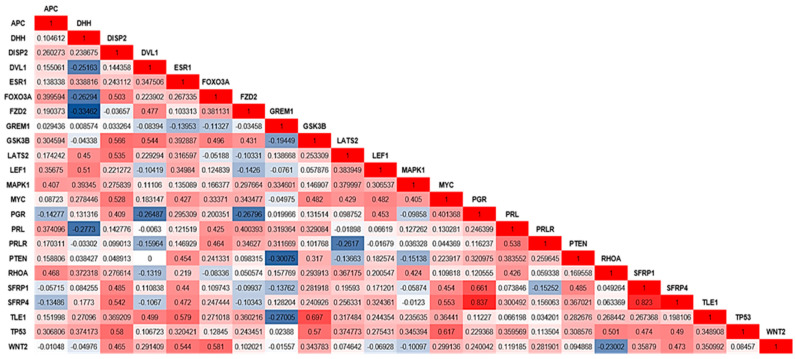
Correlation values (Spearman test) were observed for genes with significant *p*-values (*p* < 0.05 and 0.001). The color scale differentiates positive and negative correlation coefficient values: dark blue indicates a negative correlation, and red indicates a positive correlation. Gene names are indicated in the graph, as well as the correlation coefficient values.

**Figure 4 ijms-24-02431-f004:**
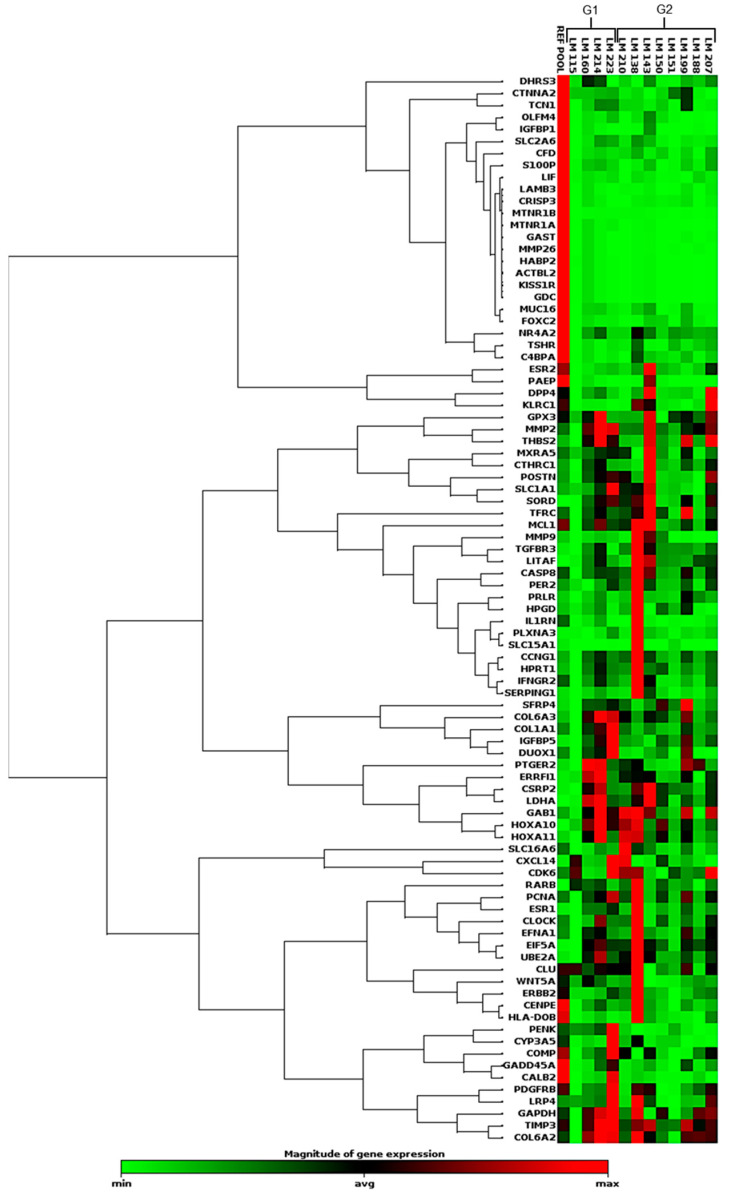
Heatmap showing the global gene expression profile of the 89 sequences evaluated by qRT-PCR, using the array-plate SYBR-green-based method (Qiagen). Red indicates upregulation, and green indicates downregulation.

**Figure 5 ijms-24-02431-f005:**
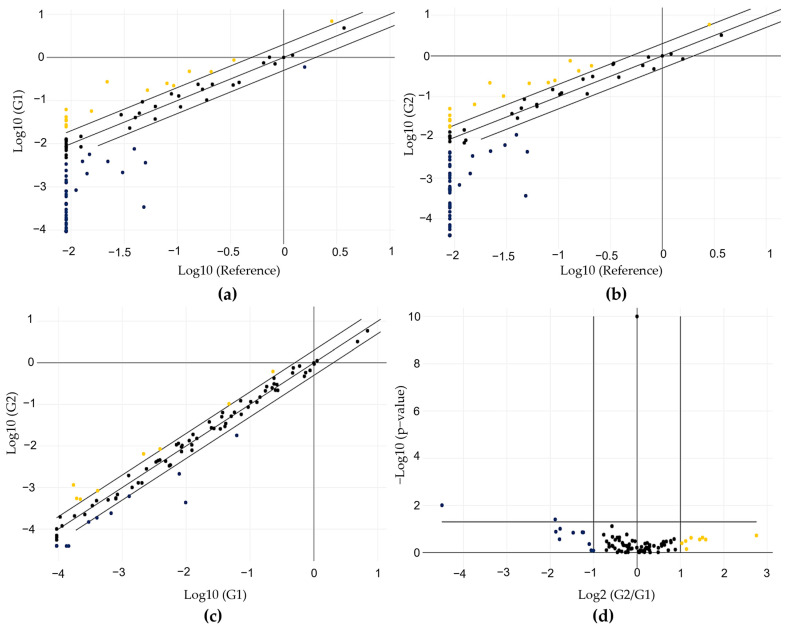
Gene expression profile of the 89 genes evaluated using the array-plate SYBR-green-based method (Qiagen). (**a**) Scatter plot showing the profile obtained from the comparison between G1 and RG; (**b**) the profile obtained from the comparison between G2 and RG; (**c**) the profile obtained from the comparison between G2 and G1; (**d**) volcano plot showing the profile obtained from the comparison between G2 and G1, considering the significance values. Yellow indicates upregulation, while blue indicates downregulation.

**Figure 6 ijms-24-02431-f006:**
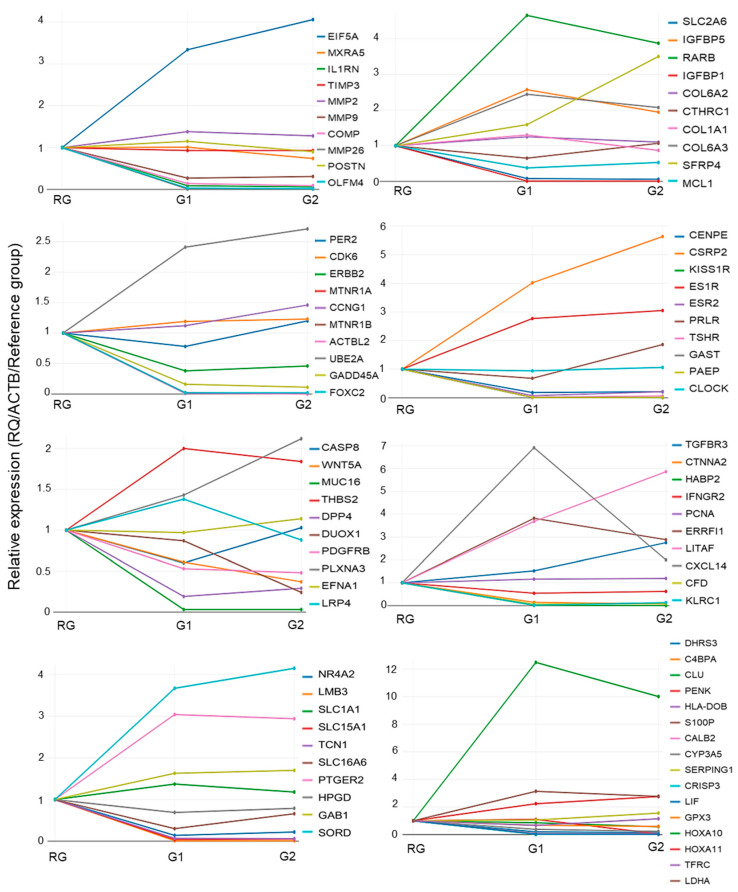
Global relative quantification values (RQs) from the 86 sequences between the groups. The values were plotted as relative gene expression values using RG as a reference value (=1). Values > 1 indicate upregulation, and values < 1 downregulation. Gene symbols and colors are indicated in the graphs.

**Table 1 ijms-24-02431-t001:** Environmental data for the metropolitan area of Sao Paulo over the last 18 years (CETESB).

Parameters	Time ^a^	Standard ^b^	Limit ^c^	Average ± SD	Range ^d^
Total inhalable material (ug/m^3^) ^e^	24 h	240	625 ^1^	-	-
	GAA	80	-	-	-
PM_10_ (ug/m^3^)	24 h	120	420	-	23–64 ^2^
	AAA	40	-	35.5 ± 6.15	27–51
PM_2.5_ (ug/m^3^)	24 h	60	210	-	-
	AAA	20	-		10–25 ^2^
Ozone (ug/m^3^)	8 h	140	400	-	-
	AAA	-	-		130–163 ^2^
Smoke (ug/m^3^)	24 h	120	420 ^1^	-	-
	AAA	40	-	28.2 ± 8.15	14–47
Carbone monoxide (ppm)	8 h	9	30		1.0–3.2 ^2^

^a^ Average calculated for 8 or 24 h, arithmetic annual average (AAA) or geometric annual average (GAA); ^b^ since 2013; ^c^ maximum limit for alert; ^d^ minimum and maximum values described in the period; ^e^ no data available for 2007, because these materials started to be presented as PM_10_, PM_2.5_, smoke, and other gases; ^1^ the reference value was recommended by 2013 legislation (National Environment Council—CONAMA); ^2^ floating average.

**Table 2 ijms-24-02431-t002:** Comparison of clinical features of G1 and G2.

Variable	n	G1	G2	*p*-Value *
Age (years)	24	42.10 ± 6.54	41.79 ± 7.07	0.9129
Menarche (years)	23	13.40 ± 2.59	12.92 ± 1.38	0.9740
IMC	24	29.55 ± 6.67	26.54 ± 5.08	0.2217
Uterine volume	22	860.1 ± 792.3	499 ± 259.2	0.1389

* Mann–Whitney test.

**Table 3 ijms-24-02431-t003:** Clinical features of the evaluated groups (n = 24) *.

Variable	Category	G1 ** N (%)	G2 *** N (%)	Total N (%)	*p*-Value
Age (years)	≤40	4 (44)	5 (56)	9 (37.5)	1.000 ^a^
	>40	6 (40)	9 (60)	15 (62.5)	
	Total	10 (42)	14 (58)	24 (100)	
Menarche (years)	≤12	3 (43)	4 (57)	7 (30)	1.000 ^a^
	>12	7 (44)	9 (56)	16 (70)	
	Total	10 (43)	13 (57)	23 (100)	
Smoking	No	6 (37.5)	10 (62.5)	16 (67)	0.891 ^b^
	Ex	3 (60)	2 (40)	5 (21)	
	Yes	1 (33)	2 (67)	3 (12)	
	Total	10 (42)	14 (58)	24 (100)	
Ethnicity	Caucasian	9 (50)	9 (50)	18 (75)	0.340 ^a^
	Afro-descendant	1 (17)	5 (83)	6 (25)	
	Total	10 (42)	14 (58)	24 (100)	
Weight (Kg)	≤60	3 (43)	4 (57)	7 (29)	1.000 ^a^
	>60	7 (41)	10 (59)	17 (71)	
	Total	10 (42)	14 (58)	24 (100)	
BMI	≤24.9	3 (30)	7 (70)	10 (42)	0.421 ^a^
	>24.9	7 (50)	7 (50)	14 (58)	
	Total	10 (42)	14 (58)	24 (100)	
Uterine volume	≤646 cm^3^	4 (29)	10 (71)	14 (64)	0.187 ^a^
	>646 cm^3^	5 (62.5)	3 (37.5)	8 (36)	
	Total	9 (41)	13 (59)	22 (100)	
Pregnancy	0	3 (33)	6 (67)	9 (37.5)	0.678 ^a^
	≥1	7 (47)	8 (53)	15 (62.5)	
	Total	10 (42)	14 (58)	24 (100)	
Parity	0	3 (33)	6 (67)	9 (37.5)	0.678 ^a^
	≥1	7 (47)	8 (53)	15 (62.5)	
	Total	10 (42)	14 (58)	24 (100)	
Abortion	No	9 (43)	12 (57)	21 (87.5)	1.000 ^a^
	Yes	1 (33)	2 (67) ^d^	3 (12.5)	
	Total	10 (42)	14 (58)	24 (100)	
Contraceptive use	No	1 (12.5)	7 (87.5)	8 (33)	0.079 ^a^
	Yes	9 (56)	7 (44)	16 (67)	
	Total	10 (42)	14 (58)	24 (100)	
LM treatment	No	8 (57)	6 (43)	14 (58)	0.104 ^a^
	Yes	2 (20)	8 (80)	10 (42)	
	Total	10 (42)	14 (58)	24 (100)	
Surgery	Myomectomy	2 (33)	4 (67)	6 (25)	1.000 ^a^
	Hysterectomy	8 (44)	10 (56)	18 (75)	
	Total	10 (42)	14 (58)	24 (100)	
Associated pathologies ^c^	No	7 (39)	11 (61)	18 (75)	0.665 ^a^
	Yes	3 (50)	3 (50)	6 (25)	
	Total	10 (42)	14 (58)	24 (100)	

* Control group data were not included because they pertained to five women from group 1. ** Patients who lived in the MRSP (G1); *** patients who lived in Sao Paulo city (G2); ^a^ Chi-square test; ^b^ Fisher’s exact test; ^c^ diabetes and blood hypertension; ^d^ one of these patients had experienced four episodes of abortion.

## Data Availability

All data generated or analyzed during this study are included in this article or the Appendix A.

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
