# Peer review of "Gene Expression Profile of Uterine Leiomyoma from Women Exposed to Different Air Pollution Levels in Metropolitan Cities of Sao Paulo, Brazil"

_ijms, 2023, doi:10.3390/ijms24032431_

Round 1
Reviewer 1 Report
The gene expression profile of uterine leiomyoma from areas with different degree of AP was investigated in the present article. There are several concerns and suggestions as below.
1. The thesis focused on the influence of AP to gene expression of uterine leiomyoma; however, the core contention could not be understanded from title.
2. Total 24 subjects were included in this study, and clinical features were listed. The relationship of each feature between uterine leiomyoma was not be explained. As well, there are important relationship on occupations, races, and the exposure period of outdoor working or exercise between uterine leiomyoma.
3. The reason why the MM from G1 patients were used as reference need to be further expounded.
Reviewer 2 Report
I read the research with great interest. It is certainly well conducted . It is not easy to understand for non-experts, but it is certainly quite effective in demonstrating the impact of air pollution on leiomyoma development, through the expression of mutated genes, and its association with women's infertility.
I ask colleagues to structure a conclusion that can be better translated into current daily practice and to extend the importance of the research into common life in large metropolitan areas, where pollution is more felt.
